# PROVABLE LEARNING OF CONVOLUTIONAL NEURAL NETWORKS WITH DATA DRIVEN FEATURES

## ABSTRACT

Convolutional networks (CNN) are computationally hard to learn. In practice, however, CNNs are learned successfully on natural image data. In this work, we study a semi-supervised algorithm, that learns a linear classifier over data-dependent features which were obtained from unlabeled data. We show that the algorithm provably learns CNNs, under some natural distributional assumptions. Specifically, it efficiently learns CNNs, assuming the distribution of patches in the input images has low-dimensional structure (e.g., when the patches are sampled from a low-dimensional manifold). We complement our result with a lower bound, showing that the dependence of our algorithm on the dimension of the patch distribution is essentially optimal.

## 1 INTRODUCTION

Convolutional Neural Networks (CNNs) have become a primary tool for machine learning on image data, surpassing the performance of "traditional" machine learning algorithms such as linear classifiers or kernel-SVM. On the other hand, unlike such simpler classifiers, neural networks are known to be computationally hard to learn. Namely, it has been shown that, under mild cryptographic assumptions, there is no efficient algorithm that can find, *under any distribution*, a neural network with good, or even non-trivial, test performance. This is true even if we assume that there exists a neural network that perfectly labels the distribution (Livni et al., 2014).

How can we explain this gap between the practical success and the theoretical hardness of neural networks? One possible answer is that we should introduce *distributional assumptions* into the analysis of learning algorithms. That is, while in theory we would like our algorithms to perform well under *any* possible distribution of examples, in practice "natural" distributions appear to enjoy a certain structure that traditional theoretical analysis does not take into account. Therefore, it seems that in order to give a theoretical analysis that is relevant to practice, we must find assumptions on the underlying data distributions that are both a) realistic for natural data and b) allow us to guarantee efficient learnability of neural networks.

In this work, we explore such possible distributional assumptions. Specifically, we study learning convolutional functions over data with a structure that is similar to natural images. Our main assumption is that the distribution of "important" patches in the images has a low-dimensional structure (e.g., lies on a low-dimensional manifold). Under this assumption, we analyze a semi-supervised algorithm that first constructs a representation based on an unlabeled set of examples, and then learns a linear classifier over the produced representation. Note that this is an *improper* learning algorithm for learning CNNs. Our algorithm is very similar to the semi-supervised algorithm introduced by Coates et al. (2011) which empirically has performance which is close to the performance of neural networks on several image datasets (Thiry et al., 2021). We show that our algorithm has run-time and sample complexity that depend on the covering number of the space of patches, and therefore the algorithm is efficient when the patches have low-dimensional structure. We complement this result with a lower bound, showing that such dependence on the covering number is essentially optimal, i.e., any learning algorithm must depend on this number in the same way that our algorithm does.

## 2    RELATED WORK

**Learning with Data-Dependent Representation**    Learning linear classifiers over fixed representations of the input is a very well-known and well-studied approach. Specifically, kernel methods (Shawe-Taylor et al., 2004) and random features (Rahimi et al., 2007; Rahimi & Recht, 2008) are some prominent examples of such learning methods. However, these methods are known to have their limitations, due to the fact that the representation is fixed a priori, and cannot adapt to the learning task (see for example Kamath et al. (2020); Daniely & Malach (2020); Malach et al. (2021)). Therefore, using representations that depend on the properties of the input data instead of fixing the representation in advance seems like a sensible way to improve performance. Indeed, such data-dependent representations have been explored for learning over image data. Specifically, the work of Coates et al. (2011) studies a learning algorithm for image data, which constructs a data-dependent representation in a very similar fashion to the algorithm we analyze in this work. However, Coates et al. (2011) focuses on empirical evaluation of the algorithm, while our work gives a theoretical analysis of the algorithm. A more recent work by Thiry et al. (2021) suggests a similar patch-based representation, and shows that it achieves impressive empirical performance on complex datasets such as CIFAR-10 and ImageNet. They do not perform theoretical analysis. Another related line of works study kernels constructed via random neural networks, such as the Neural Tangent Kernel (Du et al., 2019; 2018b; Arora et al., 2019; Ji & Telgarsky, 2019; Cao & Gu, 2019; Jacot et al., 2018).

**Learning Neural Networks under Distributional Assumptions**    As mentioned earlier, efficient learning of neural networks without any distributional assumptions is known to be hard (e.g., Klivans & Sherstov (2009); Livni et al. (2014)). In fact, in some cases learning neural networks under simple distributions such as Gaussian or Uniform distributions was shown to be hard for a large family of algorithms (see Diakonikolas et al. (2020)). This fact motivates finding the "right" distributional assumptions, that are both realistic (in some sense), and allow efficient learning of neural networks. To this end, various works have studied learning of feed-forward networks (Ge et al. (2018); Awasthi et al. (2021)) and convolutional networks under different distributional assumptions (Brutzkus & Globerson (2017); Oymak & Soltanolkotabi (2018); Du & Goel (2018); Malach & Shalev-Shwartz (2018); Brutzkus & Globerson (2020); Du et al. (2018a)). However, these distributional assumptions often seem far from fitting natural data distributions. Our work gives a simple characterization of data distributions that both seems reasonable for natural data, and allows efficient learning of convolutional networks.

## 3    PRELIMINARIES

We begin by describing the problem setting considered in the paper. First, we introduce our assumptions on the data distribution and the labeling function. We then describe the learning algorithm that we analyze.

### 3.1    DATA GENERATING DISTRIBUTION

We consider distributions over images, where each image is of size $d_I$. Let $\mathcal{X} \subseteq \mathbb{R}^{d_I}$ be the input space and $\mathcal{Y} = \{\pm 1\}$ be the label space. Each image contains $n$ patches, i.e. $n$ sub-images, where each sub-image is of size $d_P$. We identify each of the $n$ patches in the image with a sequence of indices. That is, we define $n$ sequences $A_1, ..., A_n$ such that for every $j$, $A_j = (i_1, \ldots, i_{d_P})$ is a sequence of indices satisfying $i_1, \ldots, i_{d_P} \subseteq [d_I]$. So, $A_j$ defines the set of indices of image pixels associated with the $j$-th patch, and we denote the $j$-th patch by $\mathbf{x}[j]$ s.t.:

$$\mathbf{x}[j] = \left(\mathbf{x}_{A_j(1)}, \ldots, \mathbf{x}_{A_j(d_P)}\right)$$

Note that we do not assume that $A_1, \ldots, A_n$ are disjoint, so the same pixel can belong to multiple patches. We now introduce our assumption on the function which labels the input distribution. Consider the following generalization of convolutional networks:

**Definition 3.1** (Generalizd Convolutional Function)**.** *Let $f : \mathbb{R}^{d_P} \to \mathbb{R}^k$ be some function. We say that $F^* : \mathcal{X} \to \mathbb{R}$ is a Generalized Convolutional Function (GCF) over $f$ if*

$$F^*(\mathbf{x}) = \sum_{i=1}^{n} \left\langle \mathbf{u}^{(i)}, f\left(\mathbf{x}[i]\right) \right\rangle \tag{1}$$

*where* $\mathbf{u}^{(1)}, \ldots, \mathbf{u}^{(n)} \in \mathbb{R}^k$.

Observe that a GCF is a generalization of a CNN in two aspects. First, we allow the function $f$ to be any arbitrary function over patches, and we do not restrict it only to be a neural network. Second, we allow the patch indices to be arbitrary (as discussed above), which allows implementing overlapping or non-overlapping convolutions, dilated convolutions and also arbitrary choice of "patches" from the image. Our main result shows an algorithm that can learn GCFs (under appropriate assumptions), and therefore can be applied to standard CNNs, as well as a broader family of functions.

Let $\mathcal{I}$ be some distribution over $\mathcal{X} \times \mathcal{Y}$, i.e. - a distribution of labeled examples. Given our definition of GCFs, we make the following assumption on $\mathcal{I}$:

**Assumption 3.2.** *There exists a GCF* $F^* : \mathcal{X} \to \mathbb{R}$ *over some function* $f : \mathbb{R}^{d_P} \to \mathbb{R}^k$ *such that:*

- *For any* $(\mathbf{x}, y) \sim \mathcal{I}$, $yF^*(\mathbf{x}) \geq 1$.

- $f$ *is* $L$-*Lipschitz, for some Lipschitz constant* $L > 0$.

- *There exists some* $M > 0$ *such that* $\|f(\mathbf{u})\| \leq M$ *for every* $\mathbf{u} \in \mathbb{R}^{d_P}$.

- *There exists some constant* $B > 0$ *such that the weights of* $F^*$ *satisfy* $\sum_{i=1}^n \left\| \mathbf{u}^{(i)} \right\| \leq B$.

Simply put, we assume that the distribution $\mathcal{I}$ is realizable (with a margin) by a GCF over a Lipschitz and bounded function.

### 3.1.1 STANDARD CNN ARCHITECTURES

We now show that a distribution labeled by a standard CNN architecture, with a Lipschitz and bounded activation function, satisfies Assumption 3.2. Since a CNN is an instance of a GCF, all we need to show is a bound on the Lipschitz constant $L$ and a bound on the norm $M$ of the relevant function $f$. We show this for both deep and shallow (one hidden-layer) CNNs.

Let $\sigma$ be a 1-Lipschitz activation function. Then, if $f(\mathbf{x}) = \sigma(W\mathbf{x})$, for some $W \in \mathbb{R}^{q \times P}$, a GCF $F$ over $f$ is simply a one hidden-layer CNN. In this case, we have the following result:

**Theorem 3.3.** *Let* $\sigma$ *be a* 1-*Lipschitz activation s.t.* $\sup_x |\sigma(x)| \leq c$. *Then, any distribution over* $\mathcal{X} \times \mathcal{Y}$ *labeled by a one hidden-layer CNN* $F(\mathbf{x}) = \sum_{i=1}^n \left\langle \mathbf{u}^{(i)}, \sigma(W\mathbf{x}[i]) \right\rangle$ *with activation* $\sigma$ *and weights* $W \in \mathbb{R}^{k \times d_P}$; $\mathbf{u}^{(1)}, \ldots, \mathbf{u}^{(n)} \in \mathbb{R}^k$ *satisfies Assumption 3.2 with* $L = \|W\|_2$ *and* $M = c\sqrt{k}$.

The above follows immediately fromt the following Lemma:

**Lemma 3.4.** *Let* $\sigma : \mathbb{R} \to \mathbb{R}$ *be a* 1-*Lipschitz activation function. Let* $f : \mathbb{R}^{d_P} \to \mathbb{R}^k$ *be a function such that* $f(\mathbf{x}) = \sigma(W\mathbf{x})$, *for* $W \in \mathbb{R}^{k \times d_P}$. *Then,* $f$ *is* $\|W\|_2$-*Lipschitz.*

Similarly to the above, if we let $f$ be a *deep* neural network with activation $\sigma$, namely $f(\mathbf{x}) = \sigma(W^{(t)}\sigma(\ldots \sigma(W^{(1)}\mathbf{x})\ldots))$, then a GCF $F$ over $f$ is equivalent to a depth $t$ CNN [1]. In this case, we have the following result:

**Theorem 3.5.** *Let* $\sigma$ *be a* 1-*Lipschitz activation s.t.* $\sup_x |\sigma(x)| \leq c$. *Then, any distribution over* $\mathcal{X} \times \mathcal{Y}$ *labeled by a dpeth* $t$ *CNN network*

$$F(\mathbf{x}) = \sum_{i=1}^n \left\langle \mathbf{u}^{(i)}, \sigma(W^{(t)}\sigma(\ldots \sigma(W^{(1)}\mathbf{x})\ldots)) \right\rangle$$

*with activation* $\sigma$ *and weights* $W^{(1)} \in \mathbb{R}^{k \times d_P}, W^{(2)}, \ldots, W^{(t)} \in \mathbb{R}^{k \times k}$; $\mathbf{u}^{(1)}, \ldots, \mathbf{u}^{(n)} \in \mathbb{R}^k$ *satisfies Assumption 3.2 with* $L = \prod_i \left\| W^{(i)} \right\|_2$ *and* $M = c\sqrt{k}$.

This follows immediately from the following Lemma:

**Lemma 3.6.** *Let* $f = \sigma \circ W^{(t)} \circ \sigma \circ \cdots \circ \sigma \circ W^{(1)}$, *for some* 1-*Lipschitz activation function* $\sigma$. *Then,* $f$ *is* $\prod_i \left\| W^{(i)} \right\|_2$-*Lipschitz*

---

[1]Formally, $F$ will be a depth $t$ CNN if $W^{(1)}, \ldots, W^{(t)}$ are convolutions. However, since convolutions are linear operations, our definition describes a larger family of functions.

Note that the bound on the Lipshchitz constant of deep CNNs in Theorem 3.5 grows as a product of the norm of the weights. That is, if the weight matrices have norm $> 1$, then the Lipschitz constant grows exponentially with depth, which may be unsatisfactory for deep networks. However, we do note that this is only an upper bound on the Lipschitz constant of the network, and in fact we can expect that deep networks in practice have much smaller Lipschitz constatns (see Combettes & Pesquet (2020); Jordan & Dimakis (2020)).

## 3.2 THE COVERING NUMBER OF PATCH DISTRIBUTIONS

Let $\mathcal{I}$ be a distribution over $\mathcal{X} \times \mathcal{Y}$ that satisfies Assumption 3.2 with some GCF $F^*$ over $f$. We assume that the marginal distribution of $\mathcal{I}$ on $\mathcal{X}$ has a density function and we denote its support by $\mathrm{supp}(\mathcal{I})$. We denote by $P_{\mathcal{I}}$ the set of supported patches in the distribution $\mathcal{I}$ which have non-zero value under $f$. Namely, $P_{\mathcal{I}} = \{\mathbf{x}[i] \mid \mathbf{x} \in \mathrm{supp}(\mathcal{I}), \ f(\mathbf{x}[i]) \neq 0, \ 1 \leq i \leq n\}$. In cases where the distribution $\mathcal{I}$ is clear from the context, we simply use $P := P_{\mathcal{I}}$. So, $P$ is the set of patches that both have non-zero probability to appear in images sampled from $\mathcal{I}$, and which are not "ignored" by the GCF. This captures the fact that for image classification tasks, some patches of the image may be very important for determining the object class, while other patches can simply be ignored. The set $P$ is therefore exactly this set of "important" patches.

The sample complexity and the run-time of our algorithm depend on one important factor - the covering number of the set $P$. In general, an $r$-covering number is defined as follows:

**Definition 3.7.** *For a set $A$, we say that $C$ is an $r$-covering of $A$ if $A \subseteq \cup_{\mathbf{v} \in C} \mathcal{B}_r(\mathbf{v})$, where $\mathcal{B}_r(\mathbf{v})$ is the $\ell_2$ ball of radius $r$ centered around $\mathbf{v}$. The $r$-covering number of a set $A$, denoted by $N_r(A)$, is the minimal size of an $r$-covering of $A$ (Vershynin, 2018).*

So, the main ingredient in our analysis is showing that if the covering number of the set $P$ is small (for example, if the important patches lie on a low-dimensional manifold, as described in Section 4.2), then our algorithm can efficiently learn GCFs, and therefore CNNs. Notice that assuming that $N_r(P)$ is small is a *distributional assumption*, and it does not necessarily hold in general. In fact, in some cases the covering number of $P$ can be exponential in the patch size $d_P$, or even unbounded (if the distribution of patches is unbounded), in which case our bounds become vacuous. That said, we show in Section 4.3 that in general, such dependence on the covering number is unavoidable for learning GCFs.

## 3.3 LEARNING ALGORITHM

We now describe the learning algorithm analyzed in this work. The algorithm has two stages. In the first stage, which is unsupervised, a dictionary of patches is learned by clustering patches from an *unlabeled* set of examples. In the second stage, which is supervised, a linear classifier is learned over a feature-map generated using the patch dictionary. We begin by describing the patch embedding feature-map, and continue with a detailed description of the learning algorithm.

**Patch embedding.** Let $D = \{\mathbf{v}_1, ..., \mathbf{v}_l\} \subseteq \mathbb{R}^{d_P}$ be a set of patches. We refer to $D$ as a *dictionary*. We next describe how to use a dictionary to obtain a representation of a given image $\mathbf{x}$.

For a distance parameter $\tau > 0$, we define the *patch-embedding* $\phi(\cdot; D, \tau) : \mathbb{R}^{d_P} \to \{0, 1\}^l$, where for every $1 \leq i \leq l$:

$$\phi(\mathbf{u}; D, \tau)_i = \begin{cases} 1 & \|\mathbf{v}_i - \mathbf{u}\| \leq \tau \\ 0 & \text{otherwise} \end{cases} \tag{2}$$

Namely, $\phi$ maps a given patch $\mathbf{u}$ to a vector indicating all the patches in the dictionary $D$ that are $\tau$-close to $\mathbf{u}$. We use a normalized patch embedding $\overline{\phi}$ defined as:

$$\overline{\phi}(\mathbf{u}; D, \tau) = \frac{\phi(\mathbf{u}; D, \tau)}{\|\phi(\mathbf{u}; D, \tau)\|_1}$$

Using the embedding $\overline{\phi}$, we define an *input-embedding* $\Phi(\cdot; D, \tau) : \mathcal{X} \to \{0, 1\}^{l \cdot n}$:

$$\Phi(\mathbf{x}; D, \tau) = \left[ \overline{\phi}\left(\mathbf{x}[1]; D, \tau\right), \ldots, \overline{\phi}\left(\mathbf{x}[n]; D, \tau\right) \right] \tag{3}$$

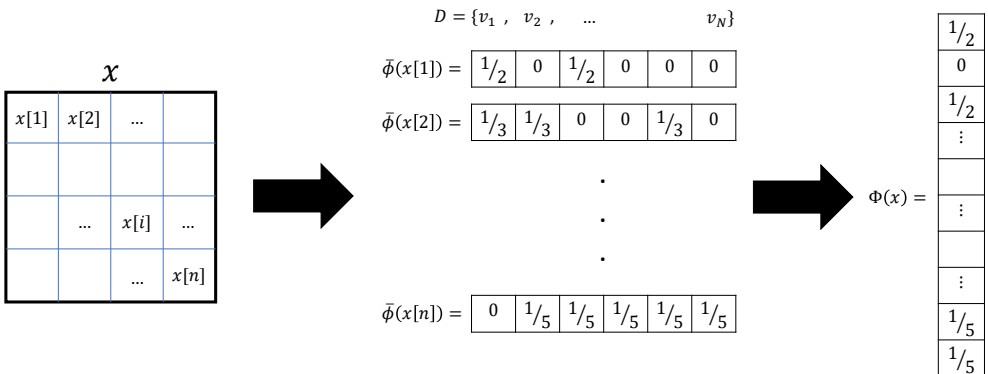

Figure 1: Construction of the patch embedding representation. First, each patch $\mathbf{x}[i]$ is mapped to a normalized indicator vector, indicating all its neighbours in a ball of radius $\tau$ within the dictionary $D$. Then the representations of all the patches are concatenated together to obtain a representation for the full image.

---

**Algorithm 1** Clustering

**Input:** Set of patches $P_u$, $N > 0$.
Pick an arbitrary $\mathbf{u} \in P_u$.
Set $D = \{\mathbf{u}\}$.
**for** $i = 2, ..., N$ **do**:
    Find $\mathbf{v} \in P_u$ which maximizes $\min_{\mathbf{u} \in D} \|\mathbf{v} - \mathbf{u}\|$.
    $D \leftarrow D \cup \{\mathbf{v}\}$
**return** $D$.

---

Figure 2: Patch clustering algorithm.

That is, $\Phi$ maps each patch of $\mathbf{x}$ using the patch embedding $\bar{\phi}$ and concatenates all patch embeddings. Figure 1 shows an illustration of how the embedding $\Phi$ is constructed using the dictionary $D$ and the input image $\mathbf{x}$.

**Learning algorithm.** We consider a semi-supervised algorithm $\mathcal{A}_{Patch}$ for learning from image data, which is similar to the algorithm presented by Coates et al. (2011). The algorithm takes two parameters: the size of the dictionary $N$, and the patch embedding radius $\tau$. The algorithm consists of an unsupervised stage, followed by a supervised stage.

Unsupervised stage: We assume that we have access to an unlabeled training dataset $S_u \subseteq \text{supp}(\mathcal{I})$. Define the set $P_u = \{\mathbf{x}[i] \mid \mathbf{x} \in S_u, 1 \le i \le n\}$, i.e., the set of all patches of images in $S_u$. We perform a clustering procedure which given $P_u$ and a number $N$, returns a set of patches $D$ of size $N$. For our theoretical analysis we will consider a greedy clustering algorithm, which performs a farthest-first traversal. The clustering algorithm is described in Figure 1. This algorithm was originally proposed for the k-center problem and it is a 2-approximation algorithm (Gonzalez, 1985).[2] In practice, other clustering algorithms can be considered, such as k-means.

Supervised stage: Here we assume that we have a dictionary $D = \{\mathbf{v}_1, ..., \mathbf{v}_N\}$ of patches obtained by the unsupervised stage. We assume that we are given a training set $S = \{(\mathbf{x}_1, y_1), ..., (\mathbf{x}_m, y_m)\}$ of labeled points, sampled IID from the distribution $\mathcal{I}$. Then, we perform a hard linear SVM on the training set $S_e = \{(\Phi(\mathbf{x}_1; D, \tau), y_1), ..., (\Phi(\mathbf{x}_m; D, \tau), y_m)\}$, and return a predictor $\bar{\mathbf{w}} \in \mathbb{R}^{N \cdot n}$ s.t.

$$\bar{\mathbf{w}} = \arg\min_{\mathbf{w}} \|\mathbf{w}\|^2 \text{ s.t. } \forall i \in [m], y_i \langle \mathbf{w}, \Phi(\mathbf{x}_i; D, \tau) \rangle \ge 1 \qquad \text{(Hard-SVM)}$$

---

[2]See also Williamson & Shmoys (2011) Section 2.2.

The prediction of the label of a new point $(\mathbf{x}, y) \sim \mathcal{I}$ is $\operatorname{sign}\left(\langle \bar{\mathbf{w}}, \Phi(\mathbf{x}; D, \tau) \rangle\right)$.

# 4 MAIN RESULT

In this section we give a detailed analysis of the algorithm presented in the previous section. Our main result shows that for every distribution $\mathcal{I}$ satisfying Assumption 3.2 with some fixed constants $L, M$ and $B$ such that the covering number satisfies $N_r(P_\mathcal{I}) \leq N$ (with some proper choice of $r$), given a large enough sample, our algorithm returns a predictor with small error.

To state this more formally, we show a learnability result, using the following notion of learnability:

**Definition 4.1.** *Let $\mathcal{P}$ be a family of distributions over $\mathcal{X} \times \mathcal{Y}$. We say that a learning algorithm $\mathcal{A}$ PAC learns the family $\mathcal{P}$, if for every $\epsilon, \delta \in (0, 1/4)$, there exists some $m(\epsilon, \delta)$ s.t. for every $\mathcal{I} \in \mathcal{D}$, the algorithm $\mathcal{A}$ uses $m(\epsilon, \delta)$ samples from $\mathcal{I}$ and returns a hypothesis $h$ such that w.p. $\geq 1 - \delta$:*

$$\mathbb{P}_{(\mathbf{x}, y) \sim \mathcal{I}}[h(\mathbf{x}) \neq y] \leq \epsilon$$

So, let $\mathcal{P}_{L,M,B,N,r}$ be the family of distributions satisfying Assumption 3.2 with constants $L, M, B$ and for which $N_r(P_\mathcal{I}) \leq N$. Then, we show that $\mathcal{A}_{Patch}$ PAC learns the family $\mathcal{P}_{L,M,B,N,r}$, for any $r \leq 1/(24LB)$. Note that this definition only addresses the supervised stage, and we will show it holds under a condition on the unsupervised stage.

We start by bounding the sample complexity of our algorithm assuming that our unlabeled training set (the data used for the unsupervised stage) is sufficient for covering the patch set $P$. Next, in Section 4.1, we show that this assumption is essentially satisfied when our unlabeled training set is large enough, and we give a bound on the required sample size of the labeled as well as the unlabeled training data (without the additional assumption).

So, we start by making the following assumption:

**Assumption 4.2.** *There exists some $r \leq \frac{1}{24LB}$, and some set $C$ s.t. $C$ is an $r$-covering of $P$ of size $N_r(P)$, and for each $\mathbf{v} \in C$, there exists $\mathbf{u} \in P_u$, such that $\mathbf{u} \in \mathcal{B}_r(\mathbf{v})$.*

That is, we assume that the unlabeled training set is representative in the sense that the patches in the unlabeled set (denoted $P_u$) cover the "important" patches of the distribution (i.e., the set $P$). Under this assumption (and the realizability assumption introduced in the previous section), we prove that our algorithm PAC learns the family of distributions $\mathcal{P}_{L,M,B,N,1/(24LB)}$, via the following theorem:

**Theorem 4.3.** *Fix $\delta \in (0, 1/4)$. Let $\mathcal{I}$ be some distribution which satisfies Assumption 3.2, and assume that $S_u$ satisfies Assumption 4.2. Then, running algorithm $\mathcal{A}_{Patch}$ with parameters $N = N_r(P)$ and $\tau = 6r$ returns a hypothesis $h$ s.t. with probability at least $1 - \delta$:*

$$\mathbb{P}_{(\mathbf{x}, y) \sim \mathcal{I}}[h(\mathbf{x}) \neq y] \leq \alpha \sqrt{\frac{nM^2 B^2 N}{m}} + \alpha \sqrt{\frac{\log(\frac{1}{\delta})}{m}} \tag{4}$$

*for some universal constant $\alpha > 0$.*

We give the full proof of the theorem in Appendix B, and give a sketch of the argument here. To prove Theorem 4.3, we show that the function $F^*$ can be approximated by a linear classifier over the representation $\Phi$. Specifically, we use the fact that the set of patches from the unlabeled data $P_u$ covers the set $P$ of patches with non-zero probability and non-zero value under $f$. So, after clustering, we still get a good covering. Next, we define the following vector:

$$\hat{\mathbf{w}} = \left(\left\langle \mathbf{u}^{(1)}, f(\mathbf{v}_1) \right\rangle, ..., \left\langle \mathbf{u}^{(1)}, f(\mathbf{v}_l) \right\rangle, ..., \left\langle \mathbf{u}^{(k)}, f(\mathbf{v}_1) \right\rangle, ..., \left\langle \mathbf{u}^{(k)}, f(\mathbf{v}_N) \right\rangle\right) \in \mathbb{R}^{Nn}$$

where the $\mathbf{v}_i$ are the vectors in the dictionary $D$ from the unsupervised stage. Then, using the guarantees on the unsupervised stage (that the dictionary covers the set of patches), the function $\hat{F}(\mathbf{x}) = \langle \hat{\mathbf{w}}, \Phi(\mathbf{x}; D, \tau) \rangle$ can be shown to approximate $F^*$. Thus, the algorithm succeeds because it can get from the unsupervised stage vectors that cover the set of patches, which in turn allows it to produce a sufficiently rich representation for learning the distribution with a linear classifier.

The above result shows how the error of our algorithm decays with the (labeled) sample size, assuming the unlabeled training set is "good". This immediately gives a bound on the sample complexity of the algorithm:

**Corollary 4.4.** *Fix $L, M, B, N > 0$ and let $r \leq \frac{1}{24LB}$. Then, $\mathcal{A}_{Patch}$ with parameters $N$ and $\tau = 6r$, given access to a sample $S_u$ satisfying Assumption 4.2, PAC learns $\mathcal{P}_{L,M,B,N,r}$ with sample complexity:*

$$m(\epsilon, \delta) = \alpha^2 \frac{nM^2B^2N + \log(1/\delta)}{\epsilon^2}$$

In Section 4.3, we show that the dependence of the sample complexity on $nN_r(P)$ is optimal.

As for the run-time of our algorithm, notice that both the unsupervised and the supervised stage use efficient (polynomial time) algorithms. The run-time of the clustering algorithm used in the unsupervised stage is $O(N^2 d_P)$, and the hard SVM used in the supervised stage runs in time $O(mn^2N^2)$, depending on the implementation.

## 4.1 Relaxing the Assumption on Unsupervised Learning

In this section we show a learning guarantee which does not require Assumption 4.2. We denote $m_u = |S_u|$. So, we show that if $m_u$ is large enough, then Assumption 4.2 is essentially satisfied with high probability:

**Theorem 4.5.** *Let $\epsilon', \delta, \delta' \in (0, 1)$. Fix $r \leq \frac{1}{24LB}$ and let $N = N_r(P)$. Assume that $m_u \geq \frac{N}{\epsilon'} \log\left(\frac{N}{\delta'}\right)$ and $m \leq \frac{\delta}{2\epsilon'}$. Then, running algorithm $\mathcal{A}_{Patch}$ with parameters $N$ and $\tau = 6r$ returns a hypothesis $h$ s.t. with probability at least $1 - \delta' - 2\delta^3$:*

$$\mathbb{P}_{(\mathbf{x},y)\sim\mathcal{I}}[h(\mathbf{x}) \neq y] \leq \alpha\sqrt{\frac{nM^2B^2N}{m}} + \alpha\sqrt{\frac{\log(\frac{1}{\delta})}{m}} + \epsilon' \tag{5}$$

*for some universal constant $\alpha$.*

The proof of Theorem 4.5 is very similar to the proof of Theorem 4.3, except that now we do not assume that we have an unlabeled set that covers the patch space. Rather, we prove that if we sample enough unlabeled examples from the marginal distribution over the inputs, such "good" data is guaranteed with high probability. This is done by observing all the balls from the covering of $P$ that have at least $\epsilon$ distributional mass, and showing that a sample from each of them is guaranteed with high probability. The full proof of the theorem is given in Appendix B.

Using the above result, we can prove a learnability result, which does not depend on Assumption 4.2. To do so, we use a more refined version of PAC learning, which distinguishes between labeled and unlabeled sample complexity:

**Definition 4.6.** *Let $\mathcal{P}$ be a family of distributions over $\mathcal{X} \times \mathcal{Y}$. We say that a learning algorithm $\mathcal{A}$ PAC learns the family $\mathcal{P}$, if for every $\epsilon, \delta \in (0, 1/4)$, there exists some $m(\epsilon, \delta)$ and $m_u(\epsilon, \delta)$ s.t. for every $\mathcal{I} \in \mathcal{D}$, the algorithm $\mathcal{A}$ uses $m(\epsilon, \delta)$ labeled samples and $m_u(\epsilon, \delta)$ unlabeled samples from $\mathcal{I}$ and returns a hypothesis $h$ such that w.p. $\geq 1 - \delta$:*

$$\mathbb{P}_{(\mathbf{x},y)\sim\mathcal{I}}[h(\mathbf{x}) \neq y] \leq \epsilon$$

Note that the two notions of PAC learnability (of Definition 4.1 and 4.6) are equivalent, as we can always use some labeled examples and disregard their labels, or otherwise not use unlabeled data at all. However, the later definition accounts separately for labeled and unlabeled sample complexity, which often can be of interest. So, Theorem 4.5 implies the following learnability result:

**Corollary 4.7.** *Fix $L, M, B, N > 0$ and let $r \leq \frac{1}{24LB}$. Then, $\mathcal{A}_{Patch}$ with parameters $N$ and $\tau = 6r$, given access to a sample $S_u$ satisfying Assumption 4.2, PAC learns $\mathcal{P}_{L,M,B,N,\tau}$ with sample complexity:*

$$m(\epsilon, \delta) = 4\alpha^2 \frac{nM^2B^2N + \log(3/\delta)}{\epsilon^2}, \quad \text{and } m_u(\epsilon, \delta) = \frac{6N \cdot m(\epsilon, \delta)}{\delta} \log\left(\frac{3N}{\delta}\right)$$

---

[3]Over the randomness of $S_u$ and $S$.

Table 1: Test classification accuracies on MNIST, Fashion MNIST (FMNIST) and noisy FMNIST.

| DATA SET | DEEP CNN | SHALLOW CNN | OUR ALGORITHM |
|---|---|---|---|
| MNIST | $99.1 \pm 0.01$ | $98.8 \pm 0.04$ | $97.7 \pm 0.09$ |
| FMNIST | $92.5 \pm 0.1$ | $90.6 \pm 0.1$ | $89.5 \pm 0.1$ |
| NOISY FMNIST | - | - | $73.1 \pm 0.1$ |

### 4.2 FROM INTRINSIC DIMENSION TO COVERING NUMBER

Our analysis, as shown in previous sections, shows that the sample complexity and run-time of our algorithm depends on one important measure - the covering number of the patch set $P$. Observe that the covering number is often used as a measure of the intrinsic dimension of some given space. For example, the $d$-dimensional ball $\mathcal{B}^d$ has covering number $N_r(\mathcal{B}^d) = C_d(1/r)^d$, and note that this remains true even if the ball is embedded in a space with larger extrinsic dimension. More generally, a bounded $d$-dimensional manifold has a covering number that grows exponentially with the intrinsic dimension $d$ (which again can be much smaller than the extrinsic dimension). For more examples and further discussion on the relation between the covering number and measures of intrinsic dimension refer to Falconer (2004); Hamm & Steinwart (2020).

All in all, we see that if the distribution of "important" patches (captured by the set $P$) is concentrated on a low-dimensional structure (e.g., on a low-dimensional manifold), we can expect the covering number of $P$ to be moderate. On the other hand, if the patches fill a truly high-dimensional space, then our complexity guarantees becomes impractical, as these can grow exponentially with the dimension. That said, we show in the next section that such dependence on the covering number is essentially unavoidable, if one wishes to learn GCFs to small loss.

### 4.3 SAMPLE COMPLEXITY LOWER BOUND

In the previous section, we showed that with sample complexity that depends on the number of patches $n$ and the covering number $N := N_r(P)$, our algorithm returns with high probability a hypothesis with small loss. Now, we will show that a dependence on $n \cdot N$ is unavoidable for guaranteeing such PAC learnability result. Namely, any algorithm that PAC learns all distribution with covering number $N$, must have sample complexity of $\Omega(nN)$:

**Theorem 4.8.** *Let $\mathcal{A}$ be some learning algorithm that uses $m$ samples. Assume that for every $\mathcal{I}$ which satisfies Assumption 3.2 and for which $N_{1/L}(P_{\mathcal{I}}) = N$, $\mathcal{A}$ returns w.p. at least $7/8$ a hypothesis $h$ s.t. $\mathbb{P}_{(\mathbf{x},y)\sim\mathcal{I}}[h(\mathbf{x}) \neq y] \leq 1/8$. Then, $m \geq nN/2$.*

The proof of Theorem 4.8 follows the same arguments as standard sample complexity lower bounds, e.g. the No-Free-Lunch Theorem (for example, see Section 5 in Shalev-Shwartz & Ben-David (2014)). The full proof is given in Appendix B.

From the above result we can derive a lower bound on the sample complexity of any algorithm that PAC learns $\mathcal{P}_{L,M,B,N,1/L}$:

**Corollary 4.9.** *Fix some $L, M, B, N > 0$. Then, for any algorithm $\mathcal{A}$ that PAC learns $\mathcal{P}_{L,M,B,N,1/L}$, it must hold that for all $\epsilon, \delta \in (0, 1/8)$:*

$$m(\epsilon, \delta) \geq \frac{nN}{2}$$

## 5 EXPERIMENTS

In this section we perform experiments to show the efficacy of our method in practice and obtain insights from our theoretical results. Coates et al. (2011) and Thiry et al. (2021) have performed extensive experiments on algorithms similar to ours. They showed that the performance of their algorithm is close to the performance of neural networks on challenging computer vision datasets. For completeness, we perform experiments on MNIST (LeCun, 1998) and Fashion MNIST (FMNIST) (Xiao et al., 2017). We compare the performance of our algorithm with a shallow CNN (2 layers) and

a deeper CNN (4 layers). We implement the clustering stage of our algorithm with k-means after performing whitening of the patches, similar to Coates et al. (2011). We used the whole training set for the unsupervised stage. See Section C for further details on the experiments. Table 1 shows the results. We see that for both MNIST and FMNIST, the semi-supervised algorithm has comparable performance to the shallow CNN and the deep CNN has the best performance.

Our theorems suggest that the sample complexity of the semi-supervised algorithm depends on the covering number of the patches. Covering numbers are related to the number of clusters of the data. If a dataset has a low covering number, it should be possible to cluster the dataset with a small number of clusters, and vice versa. Thus, by our theoretical results, we should expect that for data in which clustering algorithms work better, our algorithm should have better test performance. To check this, we plotted the mean distance to center as a function of the number of clusters for three datasets MNIST, FMNIST and a noisy version of FMNIST. Note that all patches are whitened and therefore the distances are relatively comparable between datasets. This is also illustrated by the fact that for a few clusters the mean distances to the centers are roughly the same for all datasets. The mean distance to the center measures how well the data is clustered for a given number of clusters. Figure 3 shows the results for three datasets. We see that MNIST is the most well clustered while the noisy FMNIST is the hardest to cluster. This

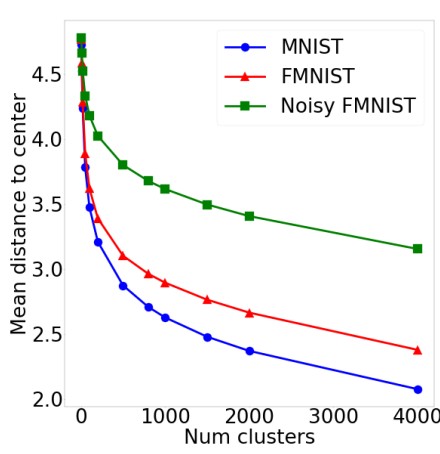

Figure 3: Mean distance to center for various number of clusters.

correlates with the test performances in Table 1, where on MNIST our algorithm has the best test performance and on noisy FMNIST it has the worse. This is in line with our theoretical results, which suggest that learning difficulty should be correlated with covering numbers.

## 6 DISCUSSION AND FUTURE WORK

In this work, we showed that although learning convolutional networks may be computationally hard in the general case, introducing distributional assumptions allows us to learn a broad class of convolutional functions efficiently. Specifically, we assumed that the "important" patches of the input have a moderate covering number (i.e., the patches are well-clustered). Under this assumption, we proved that our two step semi-supervised learning algorithm PAC learns the distribution. Following this, it is also of interest to study the behavior of other learning algorithms under the low covering number assumption. Specifically, understanding whether "vanilla" SGD on a standard CNN architecture can learn such distributions efficiently is of great interest.

Furthermore, while our assumption on the patches covering number does seem to capture some properties of natural distributions, it might still be too "generous". Since the covering number can grow exponentially with the extrinsic dimension of the patches, it might be the case that for complex datasets this number is still too large to guarantee "efficient" learnability. So, a promising future direction is to find distributional properties that lead to similar learnablility results, and yet are expected to behave more moderately on natural data. Another possible direction for research is to find distributional properties that use the specific structure of CNNs more explicitly. For example, CNNs with ReLU activation induce a separation of the patch space to linear regions. So, we can use a different sort of covering, where each linear region has at least one representative patch, such that all neighbouring patches can be associated with this linear region. Finally, another direction for future work is to explore similar distributional properties for learning other neural network architectures, such as RNNs and transformers.

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

## A    PROOFS OF SECTION 3.1.1

*Proof of Lemma 3.4.* let $x_1, x_2 \in \mathbb{R}^{d_P}$, and we have:

$$\|f(\mathbf{x}_1) - f(\mathbf{x}_2)\|^2 = \sum_i (\sigma(W\mathbf{x}_1)_i - \sigma(W\mathbf{x}_2)_i)^2$$

$$\leq \sum_i (W\mathbf{x}_1 - W\mathbf{x}_2)_i^2 = \|W(\mathbf{x}_1 - \mathbf{x}_2)\|^2 \leq \|W\|^2 \|\mathbf{x}_1 - \mathbf{x}_2\|^2$$

$\square$

*Proof of Lemma 3.6.* Observe that if $g_1$ is an $L_1$-Lipschitz function, and $g_2$ is an $L_2$-Lipschitz function, then $g_1 \circ g_2$ is $L_1 L_2$-Lipschitz:

$$\|(g_1 \circ g_2)(\mathbf{x}_1) - (g_1 \circ g_2)(\mathbf{x}_2)\| \leq L_1 \|g_2(\mathbf{x}_1) - g_2(\mathbf{x}_2)\| \leq L_1 L_2 \|\mathbf{x}_1 - \mathbf{x}_2\|$$

Now, using Lemma 3.4 gives the required. $\square$

## B    PROOF OF SECTION 4

*Proof of Theorem 4.3.* By Assumption 4.2, there exists a set $A \subseteq P_u$ of size $N_r(P)$ such that:

$$P \subseteq \cup_{\mathbf{v} \in C} \mathcal{B}_r(\mathbf{v}) \subseteq \cup_{\mathbf{v} \in A} \mathcal{B}_{2r}(\mathbf{v}) \tag{6}$$

The clustering algorithm guarantees that (Gonzalez, 1985),

$$\max_{\mathbf{v} \in P_u} \min_{\mathbf{u} \in D} \|\mathbf{v} - \mathbf{u}\| \leq 2 \max_{\mathbf{v} \in P_u} \min_{\mathbf{u} \in A} \|\mathbf{v} - \mathbf{u}\| \leq 4r \tag{7}$$

where the last inequality follows by Eq. (6). It follows that $P_u \subseteq \cup_{\mathbf{v} \in D} \mathcal{B}_{4r}(\mathbf{v})$ which implies that $\cup_{\mathbf{v} \in P_u} \mathcal{B}_{2r}(\mathbf{v}) \subseteq \cup_{\mathbf{v} \in D} \mathcal{B}_{6r}(\mathbf{v})$. Then, by Eq. (6) we get $P \subseteq \cup_{\mathbf{v} \in P_u} \mathcal{B}_{2r}(\mathbf{v}) \subseteq \cup_{\mathbf{v} \in D} \mathcal{B}_{6r}(\mathbf{v})$.

Now, define the vectors $\mathbf{w} = (f(\mathbf{v}_1), ..., f(\mathbf{v}_N)) \in \mathbb{R}^{Nk}$ and,

$$\hat{\mathbf{w}} = \left( \left\langle \mathbf{u}^{(1)}, f(\mathbf{v}_1) \right\rangle, ..., \left\langle \mathbf{u}^{(1)}, f(\mathbf{v}_l) \right\rangle, ..., \left\langle \mathbf{u}^{(k)}, f(\mathbf{v}_1) \right\rangle, ..., \left\langle \mathbf{u}^{(k)}, f(\mathbf{v}_N) \right\rangle \right) \in \mathbb{R}^{Nn}$$

i.e., $\hat{\mathbf{w}}_{(i-1)N+j} = \left\langle \mathbf{u}^{(i)}, f(\mathbf{v}_j) \right\rangle$ for all $1 \leq i \leq n$ and $1 \leq j \leq N$. Let $\hat{F}(\mathbf{x}) = \left\langle \hat{\mathbf{w}}, \Phi(\mathbf{x}; D, \tau) \right\rangle$ and notice that $\hat{F}(\mathbf{x}) = \sum_i \left\langle \mathbf{u}^{(i)}, \left\langle \mathbf{w}, \bar{\phi}(\mathbf{x}[i]; D, \tau) \right\rangle \right\rangle$.

For each $\mathbf{u} \in P$ define $T(\mathbf{u}) = \{ \mathbf{v} \in D : \|\mathbf{u} - \mathbf{v}\| \leq \tau \}$. Then, the following holds for every $\mathbf{u} \in P$:

$$\left\| f(\mathbf{u}) - \left\langle \mathbf{w}, \bar{\phi}(\mathbf{u}; D, \tau) \right\rangle \right\| = \left\| f(\mathbf{u}) - \frac{1}{|T(\mathbf{u})|} \sum_{\mathbf{v} \in T(\mathbf{u})} f(\mathbf{v}) \right\|$$

$$\leq \frac{1}{|T(\mathbf{u})|} \sum_{\mathbf{v} \in T(\mathbf{u})} \|f(\mathbf{u}) - f(\mathbf{v})\|$$

$$\leq \frac{1}{|T(\mathbf{u})|} \sum_{\mathbf{v} \in T(\mathbf{u})} L \|\mathbf{u} - \mathbf{v}\| \leq \tau L = 6rL$$

Therefore for all $\mathbf{x} \in \text{supp}(\mathcal{I})$:

$$\left| F^*(\mathbf{x}) - \hat{F}(\mathbf{x}) \right| = \left| \sum_i \left\langle \mathbf{u}^{(i)}, f(\mathbf{x}[i]) - \left\langle \mathbf{w}, \bar{\phi}(\mathbf{x}[i]; D, \tau) \right\rangle \right\rangle \right|$$

$$\leq \sum_i \left\| \mathbf{u}^{(i)} \right\| \left\| f(\mathbf{x}[i]) - \left\langle \mathbf{w}, \bar{\phi}(\mathbf{x}[i]; D, \tau) \right\rangle \right\|$$

$$\leq 6rL \sum_i \left\| \mathbf{u}^{(i)} \right\| < \frac{1}{2}$$

which implies that for any $(\mathbf{x}, y) \sim \mathcal{I}$, $y\hat{F}(\mathbf{x}) \geq \frac{1}{2}$. Therefore, $2\hat{\mathbf{w}}$ satisfies Eq. (Hard-SVM), and so we get that $\bar{\mathbf{w}}$ satisfies Eq. (Hard-SVM) with $\|\bar{\mathbf{w}}\| \leq 2\|\hat{\mathbf{w}}\|$.

Now, consider the ramp loss:

$$L^{\text{ramp}}(y, u) = \begin{cases} 1 & uy \leq 0 \\ 1 - uy & 0 < uy < 1 \\ 0 & uy \geq 1 \end{cases} \tag{8}$$

Then, by a standard generalization bound (Bartlett & Mendelson, 2002; Boucheron et al., 2005), there exists a constant $\alpha > 0$ such that with probability at least $1 - \delta$ over the randomness of $S$:

$$\mathbb{E}_{(\mathbf{x}, y) \sim \mathcal{I}}[L^{\text{ramp}}(y, \langle \bar{\mathbf{w}}, \Phi(\mathbf{x}; D, \tau) \rangle)] \leq \frac{\alpha \|\Phi(\mathbf{x}; D, \tau)\| \|\bar{\mathbf{w}}\|}{2\sqrt{m}} + \alpha \sqrt{\frac{\log(\frac{1}{\delta})}{m}} \tag{9}$$

Since the ramp loss upper bounds the 0-1 loss $L^{0-1}(y, u) = \mathbb{1}(uy < 0)$, and by the inequalities $\|\Phi(\mathbf{x}; D, \tau)\| \leq \sqrt{n}$ and $\|\bar{\mathbf{w}}\| \leq 2\|\hat{\mathbf{w}}\| \leq 2\sqrt{M^2 N_r(P) \sum_{i=1}^n \|\mathbf{u}^{(i)}\|_2^2} \leq 2\sqrt{M^2 N_r(P) B^2}$ we have:

$$\mathbb{P}_{(\mathbf{x}, y) \sim \mathcal{I}}[y\langle \bar{\mathbf{w}}, \Phi(\mathbf{x}; D, \tau) \rangle < 0] \leq \alpha \sqrt{\frac{nM^2 N_r(P) B^2}{m}} + \alpha \sqrt{\frac{\log(\frac{1}{\delta})}{m}}$$

which completes the proof. $\square$

*Proof of Theorem 4.5.* Let $C$ be an $r$-covering of $P$ of minimal size, namely $|C| = N$. Let $C' \subseteq C$ be the subset of balls that have mass $\geq \frac{\epsilon'}{N}$, i.e.:

$$C' = \left\{ \mathbf{v} \in C \; : \; \mathbb{P}_{x \sim \mathcal{I}}[\exists i \text{ s.t.} \mathbf{x}[i] \in \mathcal{B}_r(\mathbf{v})] \geq \frac{\epsilon'}{N} \right\}$$

We first show that with probability $\geq 1 - \delta'$, for every $\mathbf{v} \in C'$ we have $\mathcal{B}_r(\mathbf{v}) \cap P_u \neq \emptyset$. Indeed, fix some $\mathbf{v} \in C'$, and by definition of $C'$:

$$\mathbb{P}[\mathcal{B}_r(\mathbf{v}) \cap P_u = \emptyset] \leq \left(1 - \frac{\epsilon'}{N}\right)^{m_u} \leq \exp\left(-\frac{m_u \epsilon'}{N}\right) \leq \frac{\delta'}{N}$$

The required follows from the union bound.

Now, assume that for every $\mathbf{v} \in C'$ there exists $\mathbf{u} \in P_u$ such that $\mathbf{u} \in \mathcal{B}_r(\mathbf{v})$. Therefore, we also have $\mathcal{B}_r(\mathbf{v}) \subseteq \mathcal{B}_{2r}(\mathbf{u})$, and so:

$$\cup_{\mathbf{v} \in C'} \mathcal{B}_r(\mathbf{v}) \subseteq \cup_{\mathbf{v} \in P_u} \mathcal{B}_{2r}(\mathbf{v}) \tag{10}$$

Since $P_u \subseteq P$, there exists a set $A \subseteq P_u$ of size $N_r(P)$ such that:

$$P_u \subseteq \cup_{\mathbf{v} \in A} \mathcal{B}_{2r}(\mathbf{v}) \tag{11}$$

The clustering algorithm guarantees that (Gonzalez, 1985),

$$\max_{\mathbf{v} \in P_u} \min_{\mathbf{u} \in D} \|\mathbf{v} - \mathbf{u}\| \leq 2 \max_{\mathbf{v} \in P_u} \min_{\mathbf{u} \in A} \|\mathbf{v} - \mathbf{u}\| \leq 4r \tag{12}$$

where the last inequality follows by Eq. (11). It follows that $P_u \subseteq \cup_{\mathbf{v} \in D} \mathcal{B}_{4r}(\mathbf{v})$ which implies that $\cup_{\mathbf{v} \in P_u} \mathcal{B}_{2r}(\mathbf{v}) \subseteq \cup_{\mathbf{v} \in D} \mathcal{B}_{6r}(\mathbf{v})$. Then, by Eq. (10) we get:

$$\cup_{\mathbf{v} \in C'} \mathcal{B}_r(\mathbf{v}) \subseteq \cup_{\mathbf{v} \in D} \mathcal{B}_{6r}(\mathbf{v}) \tag{13}$$

Now, define $\hat{\mathbf{w}}$ and $\hat{F}$ as in the proof of Theorem 4.3. We say that $\mathbf{x} \in \mathcal{X}$ is *good* if for all $1 \leq i \leq n$, $\mathbf{x}[i] \in \cup_{\mathbf{v} \in C'} \mathcal{B}_r(\mathbf{v})$. We say that the training set $S$ is good if for all $\mathbf{x} \in S$, $\mathbf{x}$ is good.

By the definition of $C'$, the probability that $\mathbf{x}$ is not good is at most $\sum_{\mathbf{v} \in C \setminus C'} \mathbb{P}_{\mathbf{x} \sim \mathcal{I}} [\exists i, \mathbf{x}[i] \in \mathcal{B}_r(\mathbf{v})] \leq \epsilon'$. Let $\xi_j \in \{0, 1\}$ be the random variable indicating whether the $j$-th example in $S$ is not good. Observe that $\mathbb{E} \left[ \sum_{j=1}^m \xi_j \right] = \sum_{j=1}^m \mathbb{E} [\xi_j] = m\epsilon'$. Therefore, by Markov's inequality we have:

$$\mathbb{P} [\exists j \in [m] \ s.t. \ \xi_j = 1] = \mathbb{P} \left[ \sum_{j=1}^m \xi_j \geq 1 \right] \leq \mathbb{E} \left[ \sum_{j=1}^m \xi_j \right] = m\epsilon' < \delta$$

Therefore, the probability that $S$ is good is at least $1 - \delta$.

For each $\mathbf{u} \in \cup_{\mathbf{v} \in C'} \mathcal{B}_r(\mathbf{v})$ define $T(\mathbf{u}) = \{\mathbf{v} \in D : \|\mathbf{u} - \mathbf{v}\| \leq 6r\}$. Then, by Eq. (13), the following holds for $\mathbf{u} \in \cup_{\mathbf{v} \in C'} \mathcal{B}_r(\mathbf{v})$:

$$\left\| f(\mathbf{u}) - \langle \mathbf{w}, \overline{\phi}(\mathbf{u}; D, 6r) \rangle \right\| = \left\| f(\mathbf{u}) - \frac{1}{|T(\mathbf{u})|} \sum_{\mathbf{v} \in T(\mathbf{u})} f(\mathbf{v}) \right\|$$

$$\leq \frac{1}{|T(\mathbf{u})|} \sum_{\mathbf{v} \in T(\mathbf{u})} \|f(\mathbf{u}) - f(\mathbf{v})\|$$

$$\leq \frac{1}{|T(\mathbf{u})|} \sum_{\mathbf{v} \in T(\mathbf{u})} L \|\mathbf{u} - \mathbf{v}\| \leq 6rL$$

Therefore for all good $\mathbf{x}$:

$$\left| F^*(\mathbf{x}) - \hat{F}(\mathbf{x}) \right| = \left| \sum_i \left\langle \mathbf{u}^{(i)}, f(\mathbf{x}[i]) - \langle \mathbf{w}, \overline{\phi}(\mathbf{x}[i]; D, \tau) \rangle \right\rangle \right|$$

$$\leq \sum_i \left\| \mathbf{u}^{(i)} \right\| \left\| f(\mathbf{x}[i]) - \langle \mathbf{w}, \overline{\phi}(\mathbf{x}[i]; D, \tau) \rangle \right\|$$

$$\leq 6rL \sum_i \left\| \mathbf{u}^{(i)} \right\| \leq \epsilon < \frac{1}{2}$$

which implies that for any $(\mathbf{x}, y) \sim \mathcal{I}$ where $\mathbf{x}$ is good, $y\hat{F}(\mathbf{x}) \geq \frac{1}{2}$.

For now we consider two events: (1) Event where for every $\mathbf{v} \in C'$ we have $\mathcal{B}_r(\mathbf{v}) \cap P_u \neq \emptyset$ which occurs with probability at least $1 - \delta'$ and (2) Event that $S$ is good which holds with probability at least $1 - \delta$. Now, consider the ramp loss in Eq. (8). Then, by a standard generalization bound (Bartlett & Mendelson, 2002; Boucheron et al., 2005) and considering the events above, there exists a constant $\alpha > 0$ such that with probability at least $1 - \delta' - 2\delta$ over the randomness of $S$ and $S_u$:

$$\mathbb{E}_{(\mathbf{x},y) \sim \mathcal{I}, \ \mathbf{x} \text{ is good}} [L^{\text{ramp}} (y, \langle \overline{\mathbf{w}}, \Phi(\mathbf{x}; D, \tau) \rangle)] \leq \frac{\alpha \|\Phi(\mathbf{x}; D, \tau)\| \|\overline{\mathbf{w}}\|}{2\sqrt{m}} + \alpha \sqrt{\frac{\log(\frac{1}{\delta})}{m}} \qquad (14)$$

Since the ramp loss upper bounds the 0-1 loss $L^{0-1}(y, u) = \mathbb{1}(uy < 0)$, and by the inequalities $\|\Phi(\mathbf{x}; D, \tau)\| \leq \sqrt{n}$ and $\|\overline{\mathbf{w}}\| \leq 2 \|\hat{\mathbf{w}}\| \leq 2\sqrt{M^2 N_r(P) \sum_{i=1}^n \|\mathbf{u}^{(i)}\|_2^2}$ we have:

$$\mathbb{P}_{(\mathbf{x},y) \sim \mathcal{I}, \ \mathbf{x} \text{ is good}} [y \langle \overline{\mathbf{w}}, \Phi(\mathbf{x}; D, \tau) \rangle < 0] \leq \alpha \sqrt{\frac{nM^2 N_r(P) \sum_{i=1}^n \|\mathbf{u}^{(i)}\|_2^2}{m}} + \alpha \sqrt{\frac{\log(\frac{1}{\delta})}{m}}$$

Therefore,

$$\mathbb{P}_{(\mathbf{x},y) \sim \mathcal{I}} [y \langle \overline{\mathbf{w}}, \Phi(\mathbf{x}; D, \tau) \rangle < 0] = \mathbb{P}_{(\mathbf{x},y) \sim \mathcal{I}, \ \mathbf{x} \text{ is good}} [y \langle \overline{\mathbf{w}}, \Phi(\mathbf{x}; D, \tau) \rangle < 0]$$

$$+ \mathbb{P}_{(\mathbf{x},y) \sim \mathcal{I}, \ \mathbf{x} \text{ is not good}} [y \langle \overline{\mathbf{w}}, \Phi(\mathbf{x}; D, \tau) \rangle < 0]$$

$$\leq \alpha \sqrt{\frac{nM^2 N_r(P) \sum_{i=1}^n \|\mathbf{u}^{(i)}\|_2^2}{m}} + \alpha \sqrt{\frac{\log(\frac{1}{\delta})}{m}} + \epsilon'$$

which concludes the proof.

$\square$

*Proof of Theorem 4.8.* Assume that $m < nN/2$. For every $i \in [n]$ and $j \in [N]$, let $\mathbf{x}_{i,j} \in \mathbb{R}^{d_I}$ such that $\mathbf{x}_{i,j} = (j/L)e_k$ where $k = A_i(1)$.

Let $Z = \{\mathbf{z}_0, \mathbf{z}_1, \ldots, \mathbf{z}_N\} \subseteq \mathbb{R}^{d_P}$ a set of $N+1$ points s.t. $\mathbf{z}_i = (i/L, 0, \ldots, 0)$. Let $f : \mathbb{R}^{d_P} \to \mathbb{R}^N$ the function such that:

$$f(\mathbf{z})_j = \begin{cases} 0 & \mathbf{z}(1) \notin [(j-1)/L, (j+1)/L] \\ L \cdot \mathbf{z}(1) - j + 1 & \mathbf{z}(1) \in [(j-1)/L, jL] \\ j - L \cdot \mathbf{z}(1) + 1 & \mathbf{z}(1) \in [j/L, (j+1)L] \end{cases}$$

Namely, $f(\mathbf{z}_i) = \mathbf{e}_i$ if $i > 0$ and $f(\mathbf{z}_0) = 0$. Observe that $f$ is $L$-Lipschitz and $\|f(\mathbf{x})\| \leq 1$ for all $\mathbf{x}$. Now, for some $F(\mathbf{x}) = \sum_{i=1}^n \langle \mathbf{u}^{(i)}, f(\mathbf{x}[i]) \rangle$, notice that $F(\mathbf{x}_{i,j}) = \mathbf{u}_j^{(i)} =: y_{i,j}$ for all $i \in [n], j \in [N]$. So, for any choice of $\mathbf{y} = \{y_{i,j}\}_{i \in [n], j \in [N]} \subset \{\pm 1\}$, define the distribution $\mathcal{I}_\mathbf{y}$ defined by $(\mathbf{x}_{i,j}, y_{i,j})$ where $i \sim [n]$ and $j \sim [N]$ uniformly.

Note that $\text{supp}(\mathcal{I})$ contains $nM$ examples, and since $m < nN/2$ there are at least $nN/2$ examples in $\text{supp}(\mathcal{I})$ that are not seen by the algorithm $\mathcal{A}$. Let $S$ be the sample seen by the algorithm $\mathcal{A}$, and $\mathcal{A}(S)$ be the hypothesis returned by $\mathcal{A}$ upon seeing the sample $S$. Let $\overline{S}$ be the samples not seen by the algorithm. We also denote $\mathbf{y}(S)$ the coordinates of $\mathbf{y}$ that appear in $S$, and $\mathbf{y}(\overline{S})$ the coordinates of $\mathbf{y}$ that do not appear in $S$. So, we have:

$$\underset{\mathbf{y}}{\mathbb{E}} \, \underset{S}{\mathbb{E}} \, \mathbb{P}_{(\mathbf{x}_{i,j}, y_{i,j}) \sim \mathcal{I}_\mathbf{y}} \left[ \mathcal{A}(S)(\mathbf{x}_{i,j}) \neq y_{i,j} \right] \geq \underset{\mathbf{y}}{\mathbb{E}} \, \underset{S}{\mathbb{E}} \, \frac{1}{nN} \sum_{(\mathbf{x}, y) \in \overline{S}} \mathbf{1}\{\mathcal{A}(S)(\mathbf{x}) \neq y\}$$

$$\geq \frac{|\overline{S}|}{2nN} \geq \frac{1}{4}$$

Therefore, there exists a distribution $\mathcal{I}_\mathbf{y}$ such that

$$\underset{S}{\mathbb{E}} \, \mathbb{P}_{(\mathbf{x}, y) \sim \mathcal{I}_\mathbf{y}} \left[ \mathcal{A}(S)(\mathbf{x}) \neq y \right] \geq \frac{1}{4}$$

and observe that by definition of $\mathcal{I}_\mathbf{y}$, this distribution satisfies Assumption 3.2. Now, the required follows from a simple application of Markov's inequality (see 5.5.1 in Shalev-Shwartz & Ben-David (2014)). $\square$

## C EXPERIMENTAL DETAILS

Here we provide details of the experiments performed in Section 5. All experiments were run on NVidia Titan Xp GPUs with 12GB of memory. Neural network training algorithms were implemented with PyTorch (Paszke et al., 2019). All of the empirical results can replicated in approximately 20 hours on a single Nvidia Titan Xp GPU. The shallow network consists of a convolution layer with 32 channels with kernels of size 5x5 and stride 1, 2x2 max pooling layer and fully connected layer. The deep CNN network consists of two convolution layers followed by max pooling and two fully connected layers. The first convolution layer has 32 channels with 3x3 kernels and stride 1, the second has 64 channels and kernels as in the first convolution layer. Between the fully connected layers there are 128 hidden neurons. We used dropout in both networks after the max pooling and for the deep network also between the fully connected layer. The networks were trained with Adadelta (Zeiler, 2012) with learning rate 1.0 (it performed better than learning rate 0.1). We trained both networks for 30 epochs and chose the best performing network on a validation set. For MNIST and FMNIST, the training set has 55000 points, the validation 5000 points and the test set 10000 points.

We implemented our semi-supervised algorithm where the unsupervised stage was performed with k-means on whitened patches with 100 cluster centers. For clustering, we sampled 100000 patches of size 5x5 from the training images and performed whitening. Given cluster centers, we calculated a representation for patches. In this representation, the entry for a corresponding center is 1 if the distance between the patch and the center is less than the average of the minimum distance to the

centers and the mean distance to the centers. Otherwise it is 0. We performed k-means with 4 random seeds and 100 cluster centers and obtained 4 representations which were concatenated to get the full representation of the patch. Then, we calculated the representation for each image of the training set by concatenating the representations of the 5x5 patches of the image with stride 2 between patches. A linear classifier was trained on the new representation set of the training set which consisted of 60000 data points for all three datasets in Table 1. Then, we evaluated the algorithm using the learned linear classifer on a test set of size 10000 with the new representations.

The noisy FMNIST dataset was constructed by adding to each iamge a random isotropic Gaussian vector with zero mean and standard deviation 5. For each pair of algorithm and dataset we performed 5 runs with different Numpy and PyTorch seeds.

