# OpenReview forum: "Provable Learning of Convolutional Neural Networks with Data Driven Features"
_ICLR.cc/2022/Conference — ICLR 2022 Submitted_

### Official Review · Reviewer_nV5M · 2021-10-29

**Correctness:** 1
**Technical Novelty And Significance:** 3
**Empirical Novelty And Significance:** 1
**Recommendation:** 5
**Confidence:** 5

**Main Review:**

Section 3.1.1 is not really useful, as most of the Lipschitz bounds can be found in the literature (e.g., https://arxiv.org/abs/1312.6199) - I’d prone to remove this section, also because the considered architecture is not Lipschitz (it’s not even continuous)

Basically, this work relies on the assumption that the patches (implying the training data) are low dimensional. As stated in this work (sec 4.2), this remains to be proved, at least numerically, and this is here tractable. Secondly, if experiments can be conducted on MNIST, they should also be conducted on datasets such as CIFAR-10. These are two major flaws of this work: I believe that the idea that if some data are low dimensional then it's easier to learn is not really new and specific to this method - it needs to be shown that this is actually the good underlying model of the data.


**Summary Of The Paper:**

This work studies from the PAC-Bayes perspective the learning of predictors that consists in an aggregation of patch detectors. The main principle is to claim that if a low-dimensional structure is present in the patches, then a function of the data (e.g., a function to regress) inherits this low-dimensional structure too. From my perspective, all the proofs are direct and rely on straightforward applications of the PAC-Bayes theory and the proposed classifier isn't really studied from a game-changer perspective.

**Summary Of The Review:**

This paper tries to address the difficult question of understanding CNNs through a simplified model. While this is its main focus, I believe more works needs to be done to get to this result. Indeed, if the authors could certify that their assumptions are correct and are reflected by the data, then I’d be inclined to raise my score. In other words, that the method proposed here is actually working and employs fully the structure of some (complex) data, and is not vacuous.

---

### Official Review · Reviewer_g4Er · 2021-11-02

**Correctness:** 3
**Technical Novelty And Significance:** 2
**Empirical Novelty And Significance:** Not applicable
**Recommendation:** 3
**Confidence:** 3

**Main Review:**

** Strength **
+ The paper is well-written and well organized. It is easy to read.
+ The distribution assumption, namely Assumption 3.2, is natural, and the authors have verified the realizability of such assumption if a dataset can be classified using shallow/deep CNN.
+ Theoretical analysis seems to be sound.

** Weakness **
- The proved result seems to be "underwhelming" compared to the author's claim. The learning algorithm is constrained to learn a linear classifier on top of a pre-defined patch embedding function based on patches mined from the unlabeled data. However, the learned model is not necessarily a "standard CNN". More specifically, the patch/input-embedding function $\Phi$ is not necessarily realized by convolutions.
- The authors stressed the importance of covering number in the paper. However, no theoretical analysis or empirical study has been provided on an estimate on it. In particular, the experiments in section 5 is especially underwhelming. The authors hand-wavingly explain that noisy FMNIST appears hard to learn because it has large covering number, but no estimate has been provided, making the theoretical analysis unverified.
- Also, the scaling of the proposed algorithm is questionable. The unsupervised learning mines patches from the entire patch domain, which might be costly.
- The performance of the proposed algorithm is also not convincing. On the easy MNIST and FashionMNISt dataset, it underperforms even shallow CNNs. I would suggest test the algorithm on more complicated datasets, and report the runtime.

**Summary Of The Paper:**

The authors propose a semi-supervised learning method, that learns a linear classifier over data-dependent features obtained from unlabeled data. Under some distribution assumption on the images based on their patches, the authors claim that the algorithm provably learns CNN. Experiments are conducted on MNIST and FashinMNIST aiming to verify the importance of covering number of the patches.

**Summary Of The Review:**

Based on my review above, I am leaning towards rejecting the paper based on the utility and scalability of the algorithm, and the lack of empirical study to verify the theoretical contribution of the paper.

---

### Official Review · Reviewer_WV6X · 2021-11-04

**Correctness:** 3
**Technical Novelty And Significance:** 2
**Empirical Novelty And Significance:** 2
**Recommendation:** 5
**Confidence:** 3

**Main Review:**

### Strengths
- The paper is well-referenced, well-written, and well-motivated. It studies
  several timely issues: theory for data-driven embeddings applied in deep
  learning-esque setting (following empirical results of Thiry et al.
  mentioned), learnability of CNNs on natural data distributions.
- The mathematical formalism is precisely presented and the arguments explained
  with a great combination of technical precision and intuition, making it easy
  to understand the authors' algorithm and technical claims. I believe the
  results obtained are reasonable given these arguments, although I have not
  checked the proofs in the appendix line-by-line.

### Weaknesses

I would appreciate clarification around how the authors see several modeling
aspects of their setup, relative to the stated goal of understanding
learnability of CNNs under practical data distributions. In general, I feel an
explicit discussion of some of these aspects in the paper would be helpful:
some of this material could be worked into section 3.1.1 (although the
results there are helpful to see when thinking about $M$ and $L$, I believe
they are standard and the lemmas could be placed in the appendix).
- The GCF target model is significantly more general than CNNs, while also
  missing certain aspects of practical CNN architectures. E.g. the GCF is
  presented like a "linear probe" of a convolutional network's intermediate
  features, but I do not think modern convolutional networks use bounded
  activation functions (as in assumption 3.2); the GCF does not allow for
  hierarchical interactions between intermediate features for each patch,
  implying that representing complex visual relationships in this model will
  require $k$ to be very large (and potentially $f$ to have a large Lipschitz
  constant) -- so although $f$ can be a deep CNN, the overall classifier does
  not have some standard depth benefits that one expects in practical CNNs.
  These make me have some caution when interpreting what the theoretical
  results imply about practical, deep CNNs.
- Assumption 4.2 and the later $m_u$ condition assuming i.i.d. samples from the
  marginal image distribution seem to require the algorithm to have access to
  an unlabeled training set of all possible patches one will observe (at a
  sufficient resolution).  This might be reasonable depending on the patch
  structure in practice, but it might be helpful to discuss this a bit in the
  paper, and possibly cite prior works that argue for similar points.
- The definition of $P_{\mathcal{I}}$ is distribution-specific, but the
  algorithmic results on PAC learnability in section 4 eventually require to use
  one set $S_u$ for all possible distributions: it therefore seems like what is
  being used here via Assumption 4.2 is a definition of $P$ that does not
  involve a specific function $f$. I see that definition 4.6 attempts to handle
  this issue for the later results, but Corollary 4.7 still references
  assumption 4.2, and it is not clear to me that access to iid samples from the
  marginal distribution over images (I assume this is what is being used in
  Theorem 4.5, although it is not stated, based on the subsequent discussion)
  leads to a covering of $P_{\mathcal{I}}$ here, given that the function $f$
  does not seem to be involved in the marginal distribution.

In general, it seems from section 4.2 that the authors' argument for how their
analysis relates to the stated aim of giving theory that explains how neural
networks may be efficiently learnable in practice amounts to the fact that
learning Lipschitz functions over $d$-dimensional data has a
cursed-in-dimensionality sample complexity $\exp(d)$ (whether this $d$ is the
dimension of the ambient space, as in the hardness results of Livni et al.
cited, or the dimension of a low-dimensional manifold). I am under the
impression this is a well-known fact in computational learning theory;
moreover, the literature contains numerous results *specific* to neural
networks trained with (S)GD that prove how these dependencies can be achieved:
for example [1-5] discussed below (there are also non-algorithmic results on
approximation by deep networks, e.g. [6-9] below). The insight/contribution
here would be enhanced if the authors discussed ways in which their result on
GCFs provided insights beyond the standard result in learning theory (perhaps,
as discussed below, in terms of contributions to learning theory through their
analysis of the embedding algorithm and comparisons of their result to existing
ones).

After Corollary 4.4, the authors state that their algorithm runs in polynomial time. This description may not be appropriate, given that the parameter $N$ is not a true input parameter, but rather a derived one
  from $r$ and the properties of the set of all patches $P$ (and without
  further assumptions on $P$, one cannot guarantee a priori that this is not
  exponential in $d_P$).

[6] http://arxiv.org/abs/1908.01842

[7] http://arxiv.org/abs/2008.02545

[8] http://arxiv.org/abs/1908.00695

[9] https://openreview.net/forum?id=BJ3filKll

### Related work

It would be helpful in assessing the authors' contribution if more reference
was made to prior art in the learning theory literature that might be relevant
to the authors' algorithm and its analysis. Given the amount of study that has
been devoted to sparse coding/nearest neighbor embedding/dictionary learning
procedures in the literature, it seems to me that analysis similar to the
authors might exist. If so, it would be helpful to compare the authors' rates
to the rates in these algorithms, or point out other novelties of the authors'
results; if not, it would speak well to the novelty of the authors' analysis,
which I would like to credit. In general, the related work section is helpful,
but it spends most of the time discussing learnability results for neural
networks: it seems to me that the authors' analysis is applicable to a
significantly more general bounded Lipschitz learning setting, and lacks some
nuances associated with direct learning of NNs by GD/SGD (as mentioned above),
and so it would be appropriate to discuss more general learning results to aid
in the interpretation of the contribution.

It might be relevant to point out certain prior works that have been motivated
similarly as the authors: e.g. [1-5] below. These works imagine that the data
under consideration have a low-dimensional manifold structure, and provide some
algorithmic results for training deep networks to classify them (some
asymptotic).

[1] http://dx.doi.org/10.1103/PhysRevX.10.041044

[2] https://openreview.net/forum?id=O-6Pm_d_Q-

[3] http://arxiv.org/abs/2107.14324

[4] https://arxiv.org/abs/2106.04156

[5] http://arxiv.org/abs/2006.13409


### Minor issues/corrections
- page 5 "unsupervised stage": reference should be to figure 2 or algorithm 1
- definition 4.1: script D instead of script P?
- theorem 4.5: should this reflect what is stated in the discussion below
  ($S_u$ is composed of iid samples from the marginal over inputs?)
- bottom of page 6: $\hat{w}$ equation seems indexed incorrectly (the $v$'s
  should index from $1$ to $N$ and the $u$'s should index from $1$ to $n$)




**Summary Of The Paper:**

Motivated by the empirical successes of CNNs for classification tasks on
natural image data in spite of certain known computational hardness barriers,
the authors consider a model for semi-supervised learning of CNNs under certain
distributional assumptions. They consider a classification setup given samples
$(x, y)$ from a distribution $\mathcal{I}$, assumed to be separable with a
positive margin (in population) by a "generalized convolution function" $F$
(which is a linear function of a bounded Lipschitz function acting on patches),
which only depends on certain low-dimensional projections of the images $x$ (in
vision parlance, the "image patches"). The authors study a two-stage
semi-supervised algorithm for classifying such data, originally given by Coates
(2011): the first stage takes in unlabeled data samples $x$ and creates a patch
dictionary $D$, and the second stage uses this patch dictionary $D$ together
with a training set of i.i.d.\ labeled examples from $\mathcal{I}$ to learn a
linear classifier via solving a hard margin SVM problem with features obtained
from the dictionary $D$ (the embedding is something like a nearest-neighbor
embedding: given an input image, the image is decomposed into patches, and a
one-hot vector indicating the patches in the dictionary falling within a
specified radius of each input patch is output (and then normalized by its
sum)). The authors' analysis proves that this model PAC learns this family of
distributions given a sufficiently expressive patch dictionary $D$: the sample
complexity ultimately depends on the covering number of the set of all patches
that can be observed, and the authors argue that this indicates that when the
set of all possible patches has low-dimensional structure, we avoid the
aforementioned hardness results and learn efficiently. A lower bound is
provided to prove certain dependencies on the number of patches and the
covering number is unavoidable in the PAC learning setting. Some experiments
are provided to demonstrate the algorithm on MNIST and Fashion MNIST.



**Summary Of The Review:**

This is a well-written theory paper, but I am unclear on the extent to which it accomplishes its stated goal of providing insights into how CNNs can be efficiently learned in practice relative to the existing literature (on neural network training and on learning theory), given the very general target function model and the fact that the final bounds essentially depend on a covering number for the entire space of input patches one will observe. The CNNs that can be modeled by the authors' target function model are essentially "shallow" models with a linear probe evaluation: no hierarchical structure is captured.

---

### Official Review · Reviewer_vjtz · 2021-11-05

**Correctness:** 4
**Technical Novelty And Significance:** 3
**Empirical Novelty And Significance:** 4
**Recommendation:** 5
**Confidence:** 3

**Main Review:**

The paper is satisfyingly well-written and somehow clear and easy to follow. I am not very familiar with the literature on hardness results for neural networks and I am not able to judge the novelty of this work. It seems to be well motivated with a simple tractability result with a natural assumption for PAC learnability. I have a few comments:

I don’t think the section 3.1.1 is necessary and could be replaced by a short paragraph in order to make space for more high level discussions about the assumptions and the related literature. In particular, the relation between the covering number assumption and the settings considered in previous work (low-dimensional manifolds etc).

I am also not entirely convinced that the dependence in the covering number is the right measure here. Especially, the algorithm would require to sample from every covering ball, which does not seem to be practical. Furthermore, this covering number is on all patches at the same time, and might be impractically large. I don't think the simulations are particularly convincing.

On the other hand, some function class are tractable without requiring small covering number (by for example, restricting the GCF to have all its functions in a small RKHS ball, and taking a convolutional kernel as the learning method).

**Summary Of The Paper:**

In this paper, the authors consider learning the function class of one-layer convolutional networks. While multiple results show that learning neural networks is NP-hard in general, meaning there is no efficient algorithm that is guaranteed to succeed at learning the function class of neural networks. Hence, in order to close the gap between theory and practice, and guarantee tractability, distributional assumptions should be introduced. In this work, an abstract distribution is considered, where only the covering number is assumed to be bounded. With this assumption, the authors show that a learning algorithm with an unsupervised clustering procedure creates features that are then used in a linear classifiers, learn a class of one-layer convolutional networks, with sample complexity that is tight for the number of supervised samples in terms of $nN$.

**Summary Of The Review:**

This paper  give a simple example of tractability under the natural assumption of covering number (somehow necessary for PAC learnability). Even if it is a simple example, it is not trivial that this lower bound is achieved by an efficient algorithm, and the fact that the semi-supervised method presented in this paper works is not obvious. However, the author falls short of convincing that this covering number could a feature of real dataset that ensure tractability. For these reasons, my recommendation is marginally below the acceptance. If the authors can convince me that this covering number assumption has strong theoretical or empirical backing, I would be willing to raise my recommendation!

---

### Author Response · Authors · 2021-11-22
**Thank You**

Thank you for your thoughtful reviews which should help us improve the quality of this work. We will address your main concerns in the next revision.

---

### Decision · Program_Chairs · 2022-01-20

**Decision:**

Reject

**Comment:**

This paper proposes a new distributional assumption and a new algorithm for learning convolutional neural networks. However, the reviewers reach a consensus that this paper's assumptions are not natural and may not be satisfied in real-world domains. The meta reviewer agrees and thus decides to reject the paper.